# GENERALIZED CAPSULE NETWORKS WITH TRAINABLE ROUTING PROCEDURE

## ABSTRACT

CapsNet (Capsule Network) was first proposed by Sabour et al. (2017) and later another version of CapsNet was proposed by Hinton et al. (2018). CapsNet has been proved effective in modeling spatial features with much fewer parameters. However, the routing procedures (dynamic routing and EM routing) in both papers are not well incorporated into the whole training process, and the optimal number for the routing procedure has to be found manually. We propose Generalized GapsNet (G-CapsNet) to overcome this disadvantages by incorporating the routing procedure into the optimization. We implement two versions of G-CapsNet (fully-connected and convolutional) on CAFFE (Jia et al. (2014)) and evaluate them by testing the accuracy on MNIST & CIFAR10, the robustness to white-box & black-box attack, and the generalization ability on GAN-generated synthetic images. We also explore the scalability of G-CapsNet by constructing a relatively deep G-CapsNet. The experiment shows that G-CapsNet has good generalization ability and scalability.

## 1 INTRODUCTION

According to Sabour et al. (2017), a capsule is a group of neurons whose activity vector represents the instantiation parameters of a specific type of entity such as an object or an object part. Intuitively, CapsNet can better model spatial relationship by using much fewer parameters. The experiments in Sabour et al. (2017) show some intriguing results in MNIST (LeCun & Cortes (2010)), CIFAR10 (Krizhevsky et al.) and smallNORB (LeCun et al. (2004)). However, finding the optimal routing iterations for each capsule layer is computationally expensive which limits the scalability of CapsNet. The issue becomes more serious when a CapsNet becomes deeper. To overcome this issue, we propose Generalized CapsNet. The key idea of CapsNet is incorporating the routing procedure to the overall optimization procedure. In other words, it makes the coupling coefficients trainable instead of being calculated by dynamic routing (Sabour et al. (2017)) or EM routing (Hinton et al. (2018)).

Another interesting question yet to answer is that how to package a capsule from activations of previous convolutional layers. We can select the elements of each capsule at the same position across different feature maps (as Figure 2 Left shows), or we can select the elements of each capsule within each feature map (for example, we can choose each row or column as a capsule as Figure 2 Right shows). Two methods seem different, and the previous one makes more sense since capsules are supposed to capture different spatial features. However, according to our experiment, two ways have no significant difference. We speculate that the packing happens at the same time as initialization, so the way of packing does not matter a lot since later training will enforce capsules to learn different features.

The scalability of a neural network is crucial. As a toy example, we design a 5-layer neural network whose first two layers are convolutional layers, followed by two convolutional capsule layers, and then followed by one fully-connected capsule layer. We test this structure on CIFAR10 and find that the multi-layer G-CapsNet achieves better performance compared to the baseline by using far fewer parameters (9.6M VS 716K). We believe that deeper networks can be built to get better performance, but that is beyond the discussion.

CapsNet is better at capturing the relationship of different spatial features, so it makes sense to believe that CapsNet has a better generalization ability. To verify if this is true, we use AC-GAN

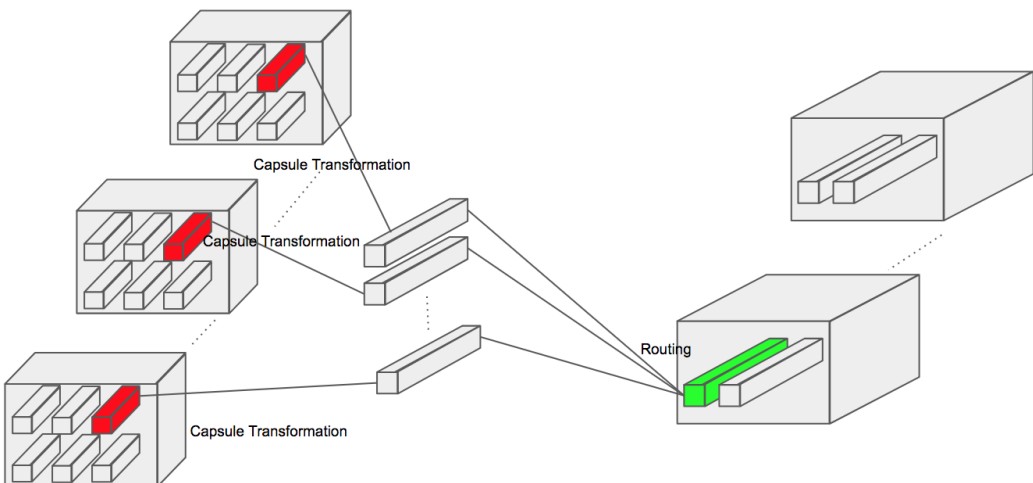

Figure 1: The structure of G-CapsNet.

proposed by Odena et al. (2017) to generate 4K synthetic images. On the one hand, the GAN-generated images are not identical to the real distribution, and on the other hand, these images are distinguishable to human beings. Both properties make these images be perfect samples to test generalization ability. The experiment shows G-CapsNet does have better generalization ability compared to CNNs.

Hinton et al. (2018) argue that CapsNet is robust to a white-box adversarial attack of FGSM Goodfellow et al. (2014). However, FGSM is not a strong attacking technique, and thus CapsNet could still be vulnerable when facing a stronger attack. Inspired by Moosavi-Dezfooli et al. (2016), we adopt a deep generative neural network to get universal perturbations offline. We found that G-CapsNet is as vulnerable as standard CNNs regarding the white-box attack. We found a similar vulnerability of G-CapsNet under black-box attack (FGSM) which is coherent with what Hinton et al. (2018) found.

## 2 RLATED WORK

The idea of CapsNets attracts much attention recently. Xi et al. (2017) explore whether stacking more capsule layers or adding more convolutional layers will result in better performance. Wang & Liu (2018) introduce a regularizer of KL divergence to minimize the clustering loss between capsules of adjacent layers. Rawlinson et al. (2018) explore how to adapt CapsNet for unsupervised learning by sparsify the last capsule layer. Jaiswal et al. (2018) adopt CapsNets as the backbone for GAN and achieve lower error rate on both MNIST and CIFAR10. O' Neill (2018) replace normal neural networks with CapsNet for face verification. We can also find some applications that are based on CapsNet. For example, Duarte et al. (2018) use CapsNet for video classification and Liu et al. (2018) adopt CapsNet for object localization.

Despite many applications and variations of CapsNets, the dynamic routing or EM routing is not well incorporated into the whole training process. For example, almost all papers mentioned above including Xi et al. (2017); Rawlinson et al. (2018); Liu et al. (2018); O' Neill (2018) etc., found that routing iterations affect the performance. The primary issue of the routing procedures in (Sabour et al. (2017); Hinton et al. (2018)) is that they are not part of the optimization procedure, we do not know the optimal number for it. The routing iterations is a meta-parameter that needs to be set manually which limits the scalability of CapsNet, especially for deep CapsNets. In the next couple of sessions, we will introduce G-CapsNet, how it works and its performance in the experimental part.

# 3 GENERALIZED CAPSNET

To better illustrate the idea of G-Capsnet, let's check what the normal neural networks' loss function looks like. Assume we have a dataset $\{(x^{(1)}, y^{(1)}), \ldots, (x^{(m)}, y^{(m)})\}$. If we can use neural networks to train a model, the loss function can be defined as Equation 1:

$$J(W, b) = \left[ \frac{1}{m} \sum_{i=1}^{m} \left( \frac{1}{2} \left\| h_{W,b}(x^{(i)}) - y^{(i)} \right\|^2 \right) \right] + \frac{\lambda}{2} \sum_{l=1}^{n_l - 1} \sum_{i=1}^{s_l} \sum_{j=1}^{s_{l+1}} \left( W_{ji}^{(l)} \right)^2 \tag{1}$$

The loss function for G-CapsNet is similar. The only difference is we have extra coupling coefficients (through routing procedure) to train. Note that $\left\| h_{W,b}(x^{(i)}) \right\|$ is the mathematical form of the neural network that we assume would fit the dataset. $n_l$, $s_l$, $s_{l+1}$ are the number of layers, the number of neurons of layer $l$ and the number of neurons of layer $l + 1$. $W$ and $b$ are the parameters we are supposed to get by training the neural network.

For the G-CapsNet, $\hat{W}_{ji}^{(l)}$ is the transformation matrix that maps one type of capsule in one layer to another type of capsule on top of it. $c_{ji}^{(l)}$ is the coupling coefficient between adjacent layers. $p_l$ and $p_{l+1}$ are the number of capsules in layer $l$ and layer $l + 1$.

$$\hat{J}(W, b, c) = \left[ \frac{1}{m} \sum_{i=1}^{m} \left( \frac{1}{2} \left\| \hat{h}_{W,b,c}(x^{(i)}) - y^{(i)} \right\|^2 \right) \right] + \frac{\lambda}{2} \sum_{l=1}^{n_l - 1} \left( \sum_{i=1}^{s_l} \sum_{j=1}^{s_{l+1}} \left( \hat{W}_{ji}^{(l)} \right)^2 + \sum_{i=1}^{p_l} \sum_{j=1}^{p_{l+1}} \left( c_{ji}^{(l)} \right)^2 \right) \tag{2}$$

## 3.1 STRUCTURE OF GENERALIZED CAPSNET

Similar to CapsNets, each capsule layer of G-CapsNet also has two operations: capsule transformation and capsule routing. As Figure 1 shows, capsule transformation does the trick of dimension transformation between adjacent capsule layers while capsule routing combines the transformed capsules.

### 3.1.1 CAPSULE TRANSFORMATION

Capsule transformation happens between adjacent capsule layers. It converse one type of capsules into another type of capsules. For example, in Sabour et al. (2017) 8-dimension capsules are transformed into 16-dimension capsules while in Hinton et al. (2018) $4 \times 4$ capsules are transformed into $4 \times 4$ capsules. In theory, we can transform type of capsules into any other type of capsules. The capsules can be of any shape (vector, matrix, cube or even hyper-cube). Assume the shape of capsules $\mathbf{u}$ in the lower layer is $(m_1, m_2, \ldots, m_k)$, the transformation matrix $\mathbf{W}$ is $m_k, n_1, n_2, \ldots, n_k$, then the shape of output capsules $\mathbf{v}$ in the higher layer is $m_1, m_2, \ldots, m_k, n_1, n_2, \ldots, n_k$. As Equation 4 shows, $\mathbf{u_i}$ is transformed to $\mathbf{u_{j|i}}$ by multiply it self with $\mathbf{W_{ij}}$.

$$\mathbf{u_{j|i}} = \mathbf{W_{ij}} \mathbf{u_i} \tag{3}$$

### 3.1.2 CAPSULE ROUTING

Capsule routing ensures capsules in lower layers are scaled and sent to their parent capsules in higher layers. From another point of view, capsule routing can be considered as a clustering procedure in which the capsules in the upper layer are centers while the capsules in the lower layer are points that need to be chosen. Capsule routing combines information to form new capsules $\mathbf{v_j}$, as Equation 4 shows.

$$\mathbf{v_j} = \sum_i c_{ij} \mathbf{u_{j|i}} \tag{4}$$

## 3.2 ACTIVATION & LOSS FUNCTION

The idea of squash function is to map the length of a capsule to a number between 0 and 1. G-CapsNet also adopts squash function as the activation function. As Equation 5 shows, $\mathbf{v_j}$ are squashed capsules and $\mathbf{v_j'}$ are capsules before the squash operation. We adopt two versions of squash functions which are proposed by Sabour et al. (2017) and Edgar et al. (2017). Our experiment shows that the difference between the two squash functions are negligible.

$$\mathbf{v_j'} = \frac{\|\mathbf{v_j}\|^2}{1 + \|\mathbf{v_j}\|^2} \frac{\mathbf{v_j}}{\|\mathbf{v_j}\|} \qquad \mathbf{v_j'} = \left(1 - \frac{1}{\mathbf{e}^{\|\mathbf{v_j}\|}}\right) \frac{\mathbf{v_j}}{\|\mathbf{v_j}\|} \tag{5}$$

For G-CapsNets, we adopt the same margin loss function in Sabour et al. (2017), as Equation 6 shows. $m^+$ is the upper threshold of the length of a capsule. In other words, capsules with a length bigger than $m^+$ are considered convergence. Similarly, $m^-$ is the lower threshold. $\lambda$ is

$$L_k = T_k \max(0, m^+ - \|\mathbf{v_k}\|)^2 + \lambda(1 - T_k) \max(0, \|\mathbf{v_k}\| - m^-)^2 \tag{6}$$

## 4 EXPERIMENTS

### 4.1 G-CAPSNET ON MNIST & CIFAR10

#### 4.1.1 FULL CONNECTED G-CAPSNET ON MNIST

We adopt the same baseline as described in paper Sabour et al. (2017) which has there convolutional layers of 256, 256, and 128 channels. The kernels and strides are 5x5 and 1. The last convolutional layer is followed by two fully-connected layers of size 328 and 192. The final layer is a 10-class softmax classifier.

As for G-CapsNet, we also adopt the same architecture as in paper Sabour et al. (2017). The first convolutional layer outputs 256 feature maps. The second convolutional layer outputs 256 feature maps or 32×6×6 8D capsules. For the final layer, we replace the dynamic routing procedure with our trainable routing procedure. Please check our released code or the original paper written by Sabour et al. (2017) for more details.

We call the capsule structure here and the one in Sabour et al. (2017) fully-connected CapsNet since each capsule in the higher layer connects to every capsule in the lower layer. As Table 1 shows, no matter whether the reconstruction involved, G-CapsNet can always achieve better performance by using much less number of parameters. Note that the performance of the baseline and the G-CapsNet reported here is a little lower than in Sabour et al. (2017), we consider the difference is caused by no pixel-shifting technology is adopted as in Sabour et al. (2017).

Another interesting thing we found is that the way of stacking the activation of capsules does not matter a lot (0.68 VS 0.66) in terms of performance. It makes more sense to package capsules across different feature maps since the capsules are supposed to capture a couple of different types of features at the same position, as Figure 2 left shows. However, what we found is that packaging capsules within each feature map (as Figure 2 right shows) as opposed to across feature maps have similar performance. It seems like no matter how we package capsules, once the organization of these capsules is fixed, the network will finally learn potential spatial relationship.

#### 4.1.2 CONVOLUTIONAL G-CAPSNET ON MNIST

Similar to the structure in Hinton et al. (2018), we also build a convolutional version of G-CapsNet. The same type of capsules of different positions share the same transformation matrices. We use a 6x6 kernel for the last capsule layer and a 4x4 "matrix" to transform capsules (from 4x4 matrix capsules to 4x4 matrix capsules). As Table 1 shows, convolutional G-CapsNet achieves better performance compared to the baseline by using fewer parameters.

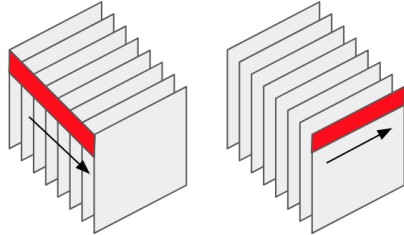

Figure 2: Packaging capsules across feature maps VS within feature maps. The red bars represent capsule vectors.

| Algorithm | error rate(%) | number of parameters |
|---|---|---|
| baseline | 0.83 | 35.4M |
| full connected G-CapsNet | 0.66 | 8.2M |
| full connected G-CapsNet* | 0.66 | 6.8M |
| full connected G-CapsNet** | 0.76 | 8.2M |
| Convolutional G-CapsNet | 0.75 | 6.9M |
| Convolutional G-CapsNet* | 0.70 | 5.5M |

Table 1: Error rate VS number of parameters on MNIST. Note that "*" means no reconstruction, "**" means the alternate way of stacking capsules.

### 4.1.3 G-CAPSNET ON CIFAR10

For CIFAR10, we build a similar G-CapsNet structure as the one used for MNIST. There are two differences. One is that we adopt 64 feature maps (other than 256 feature maps) in the first convolutional layer. The other one is we adopt 8x8 kernel for the capsule layer and transform each capsule by using an 8x8 matrix. The baseline shares the same structure as the G-CapsNet used here for the first two layers, then it has two fully-connected layers with an output of 512 and 10 separately. As Table 2 shows, both fully-connected CapsNet, and convolutional CapsNet achieve better performance than the baseline by using fewer parameters.

### 4.1.4 MULTI-LAYER G-CAPSNET ON CIFAR10

To test the scalability of G-CapsNet, we build a network with two convolutional layers, two convolutional capsule layers, and one full connected layer, as Table 3 shows. We adopt one ReLU layer and one squash layer after each convolutional capsule layer and full connected capsule layer. Take the 'Conv Caps Transform#1' as an example, (4, 4, 8) means the kernel size is (4, 4) and the number of capsule feature maps is 8. The following number is the stride. The last (8, 8) refers to the transformation matrix. Namely, the G-CapsNet transforms capsules (1x8) on the lower layer to capsules (1x8) on the upper layer. 'Conv Caps Routing#1' follows the 'Conv Caps Transform#1', whose responsibility is combining the information from each tensor with the shape of (3, 3, 16), and we have a total number of 7x7x8 combinations. For the full connected CapsNet, take the 'FC Caps Transform#1' layer as example, the multi-layer G-CapsNet transforms each tensor with a shape of (16, 3, 3, 8) on the lower layer to a new tensor with a shape of (16, 3, 3, 8) on the upper layer. Then the layer 'FC Caps Routing#1' combines each tensor of shape (16, 3, 3, 8) to a new tensor of shape (1, 8). Since CIFAR has ten classes, the output of the final layer has a shape of (10, 8). The performance of the Multi-layer CapsNet is shown at the last line of Table 2.

### 4.2 THE GENERALIZATION OF G-CAPSNET

To test the generalization ability of G-CapsNet, we adopt Auxiliary Classifier Generative Adversarial Network (AC-GAN) proposed by Odena et al. (2017) to generate 4k artificial MNIST-like images (400 images for each digit), as Figure 3 shows. AC-GAN encodes both class labels as well as the noise to generate synthetic images. The discriminator of AC-GAN also output distribution over the

| Algorithm | error rate(%) | number of parameters |
|---|---|---|
| baseline | 36.62 | 9.6M |
| full connected G-CapsNet | 32.11 | 2.67M |
| full connected G-CapsNet* | 33.00 | 2.67M |
| Convolutional G-CapsNet | 32.92 | 2.67M |
| Multi-layer G-CapsNet | 34.21 | 716K |

Table 2: Error rate VS number of parameters on CIFAR10. Note that "*" means the alternate way of stacking capsules.

| Layer | Layer parameters | Number of parameters |
|---|---|---|
| Conv#1 | (7, 7, 64), 1 | 9.4K |
| Conv#2 | (7, 7, 128), 2 | 401.4K |
| Conv Caps Transform#1 | (4, 4, 8), 1, (8, 8) | 131.1K |
| Conv Caps Routing#1 | (3, 3, 16), (7, 7, 8) | 56.4K |
| Conv Caps Transform#2 | (3, 3, 16), 2, (2, 2) | 4.6K |
| Conv Caps Routing#2 | (3, 3, 8), (3, 3, 16) | 10.4K |
| FC Caps Transform#1 | (16, 3, 3, 8), (8, 10) | 92.2K |
| FC Caps Routing#1 | (16, 3, 3, 8), 10 | 11.5K |
| Total | - | 716K |

Table 3: The structure of Multi-layer G-CapsNet on CIFAR10.

sources and the class labels. Please refer to the original paper for details. The discriminator contains four convolutional layers which have 32, 64, 128, and 256 feature maps. Each convolutional layer is followed by a leaky ReLU layer and a dropout layer. The generator first uses a fully-connected layer to map the latent vector (100, 1) to a (3, 3, 384) tensor, then up-samples the tensor to a (7, 7, 192) tensor, a (14, 14, 96) tensor and a (28, 28 ,1) tensor. The learning rate (2e-4) and beta1 (0.5) are those recommended by Radford et al. (2015).

These images are perceptible for human beings but not the same as the original images in MNIST (in terms of distribution) since they are generated in the early stage of AC-GAN. Being perceptible means these images have enough clues to tell their classes and being generated in the early stage means they do not share the same distribution as the authentic MNIST. Both properties make these images perfect for testing generalization.

The underlying assumption of G-CapsNets is that they are better at capturing the spatial features in an image and thus they need far less number of parameters than standard neural networks. As Table 4 shows, both full connected version, and convolutions version of G-CapsNet achieve better performance than the Baseline which uses much more parameters. However, both the baseline and the G-CapsNets show no significant difference when the number of epochs is larger than 3. We argue that the reason is that the generated images have already converged to the same distribution as in MNIST. Note that it makes more sense to compare the performance over three network structures other than across epochs since the generated images vary each time.

| Epochs | Baseline (35.4M) | FC G-CapsNet (6.8M) | Conv G-CapsNet (5.5M) |
|---|---|---|---|
| 1 | 19.80 | 23.50 | 19.725 |
| 2 | 5.60 | 3.25 | 4.05 |
| 3 | 0.725 | 0.00 | 0.70 |
| 4 | 0.0005 | 0.00 | 0.00 |
| 5 | 0.00275 | 0.00 | 0.001 |

Table 4: Error rate on MNIST-like images that are generated by AC-GAN. FC CapsNet is full connected G-CapsNet and Conv G-CapsNet refers to convolutional G-CapsNet.

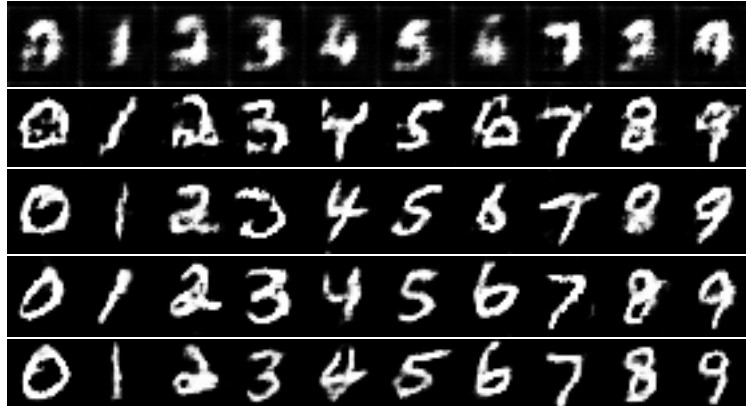

Figure 3: The generated MNIST-like images. From top to bottom, the number of training epochs is 1, 2, 3, 4 and 5.

### 4.3 THE ROBUSTNESS OF G-CAPSNET ON BLACK BOX ATTACK

Hinton et al. (2018) claimed that CapsNets are comparable to CNNs regarding robustness to black-box attacks despite using fewer parameters. Our experiment also supports this claim. Specifically speaking, we adopt LeNet as the substitute model to generate perturbations for each testing images (10K testing images in MNIST) based on FSGM (Goodfellow et al. (2014)). We restrict the maximum perturbation for each pixel to be 8 ($L_\infty \leq 8$). As Table 5 shows, the accuracy of the baseline, the fully-connected G-CapsNet, and the convolutional G-CapsNet drops sharply. There are no significant differences among them, and we can thus conclude that G-CapsNet is as vulnerable as the standard neural networks.

|  | Baseline | FC G-CapsNet (6.8M) | Conv G-CapsNet (5.5M) |
|---|---|---|---|
| accuracy | 11.35% | 8.92% | 11.35% |

Table 5: Black box attack on three types of networks.

### 4.4 THE ROBUSTNESS OF G-CAPSNET ON WHITE-BOX ATTACK

Hinton et al. (2018) also claimed that CapsNets are robust to white-box adversarial attack due to numerical instability Nayebi & Ganguli (2017) as well as the smaller percentage of zero values in the gradient. However, the whole testing is based on FGSM Goodfellow et al. (2014) which is not a strong attack technique. Inspired by Universal Adversarial Perturbation (UAP) Moosavi-Dezfooli et al. (2016), we trained a generative model to generate universal perturbations during training (offline), and then apply the generated perturbations to all testing images.

The structure of the generative model is similar to GAN but with the discriminator unchanged. Specifically, the input of the generator is a 100-dimension latent vector whose value is between 0 and 1. The latent vector layer is followed by three deconvolutional layers (note that we apply Batch Normalization after each de-convolutional layers). The final layer of the generative branch is supposed to output a tensor with the same dimension as the input image. The discriminator is the classification model we need to test. The loss function is to minimize the difference between the logits of a clean image and the logits of its manipulated version. The whole attack is untargeted, and we assign each under-attacked image a random incorrect label during training.

We apply this attack technique on the testing set of MNIST which contains 10K images. As Figure 4 shows, all three networks' accuracy drop sharply after 100 attacking iterations. This result is not consistent with what Hinton et al. (2018) found. The CapsNets and CNNs are both vulnerable to strong white-box attacks like UAP.

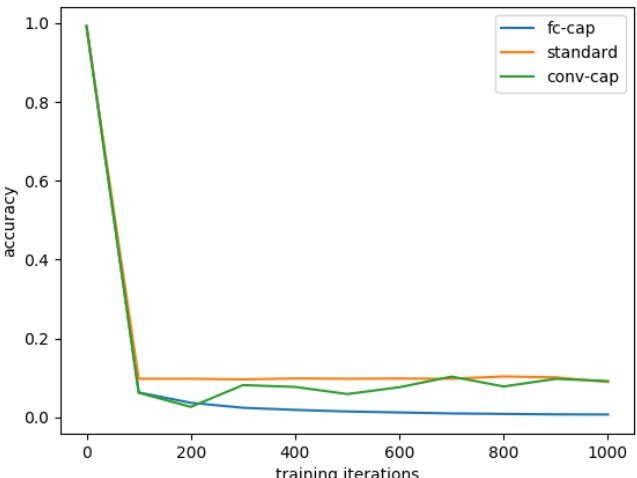

Figure 4: The decreasing curve of three network structures (standard: standard neural network, fc: fully-connected G-CapsNet, conv: convolutional G-CapsNet) with white-box attack

## 5    CONCLUSION

G-CapsNet incorporates matrix transformation and capsule routing procedures into the whole optimization process which gets rid of the set of routing times for each capsule layer and guarantees the convergence. The two versions of G-CapsNet (fully-connected G-CapsNet and convolutional G-CapsNet) achieve better performance on MNIST and CIFAR10 by using much fewer parameters. We show the good scalability of G-CapsNet by constructing a multi-layer G-CapsNet. We also use GAN-generated synthetic images to test the generalization ability of G-CapsNet. Finally, we evaluate the robustness of G-CapsNet in the background of both the white-box attack and black-box attack. For all the measurements we adopt in this paper, G-CapsNets achieve either better (concerning the accuracy, number of parameters, generalization ability, scalability) or comparable (regarding the white-box attack and black-box attack) performance.

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
