# OpenReview forum: "Generalized Capsule Networks with Trainable Routing Procedure"
_ICLR.cc/2019/Conference_

### Official Review · AnonReviewer1 · 2018-10-27
**The paper is poorly written, not clear about the contributions, evaluation is questionable**

**Rating:** 4
**Confidence:** 5

**Review:**

Pros:

The paper claims to make CapsuleNet's routing trainable. The proposed G-CapsNet have two variants (within feature map and across feature map).

It presents evaluation of G-CapsNet in terms of robustness and generalization. It is interesting that Capsule networks are as bad as traditional CNNs for strong white box attacks.

Cons:

The idea is not clearly explained. It seems that the main idea is to relax routing from a discrete problem to a continuous problem. Making the routing assignments as a regularization term in the loss function. The assignment fraction can be trained end-to-end. However, the paper discusses two loss functions in Equation 2 and 6. It is not clear how the two loss functions are related.

It is not clear why fractional routing is more efficient than the original non-trainable CapsuleNet routing. There is not clear explanation or evaluation.

The authors did not reproduce the baseline CNN model used in the original CapsuleNet routing. The original one is 0.39% error rate and the authors' implementation is 0.83%. So this makes G-CapsNet result much worse than the original CapsuleNet. So it is not clear what the benefit of G-CapsNet over the original one.

There is a related paper on approximate routing, see:
Neural Network Encapsulation, ECCV 2018
http://openaccess.thecvf.com/content_ECCV_2018/papers/Hongyang_Li_Neural_Network_Encapsulation_ECCV_2018_paper.pdf

Overall, the paper does not have enough contributions both in terms of new methods and evaluations. It does not meet the expectation of ICLR acceptance.

Response to rebuttal:
The authors clarified the questions. However, I maintain my rating because the contributions are limited and the paper is very poorly written.

---

> ### Author Response · Authors · 2018-11-06
> **Our contribution is providing a salable CapsNet.**
>
>
> Q: The relations between Equation 2 and 6.
> A: The loss function in Equation 2 shows that the convergence of G-CapsNet can be guaranteed mathematically, just like standard neural networks. In contrast, the CapsNet in (Sabour et al., 2017) can not ensure convergence mathematically since the computation of coupling (routing) coefficients is not part of the optimization. For example, the best routing number for MNIST is 3, as suggested in (Sabour et al., 2017). We found that if the routing number is 4 or larger, the performance degraded. The loss function in Equation 6 gives details. You are right that the relation between Equation 2 and 6 is not clear, we will add an explanation.
>
> Q: Explain why efficient?
> A: Thanks for pointing this out. We will add a detailed explanation as well as evaluation. The efficiency of G-CapsNet is that it does not need to set the routing number manually, and the optimal routing coefficients can be acquired, as Equation 2 shows. For example, in the paper (Sabour et al., 2017), the routing number has to be set as 3. A routing number that is smaller than 3 or larger than 3 would cause degradation of performance.  For a 10-layer CapsNet, assume we have to try 3 routing numbers for each layer, then totally we have to try 3^10 times to find the best routing number assignment. That is why the scalability and efficiency of CapsNets are questioned. For G-CapsNets, we can compute the routing coefficients just like computing the normal parameters of a neural network, and the (local) optimality can be guaranteed.
>
> Q: Why accuracy is lower?
> A: There are two reasons that our accuracy is lower than the original one. 1). We do not use any data-augmentation technique while the original paper (Sabour et al., 2017) adopts 2-pixel shifting, "Training is performed on 28 × 28 MNIST (LeCun et al. [1998]) images that have been shifted by up to 2 pixels in each direction with zero padding." 2). We use different frameworks. We use Caffe (https://github.com/chenzhenhua986/CAFFE-CapsNet) while the original paper uses TensorFlow (https://github.com/Sarasra/models/tree/master/research/capsules/).
>
> The benefit of G-CapsNet is that we can build a deep CapsNet easily without worrying about divergence. For CapsNets (Sabour et al., 2017), finding the best routing number for a deep network would be computationally expensive. Our contribution is providing a scalable version of CapsNet while all other advantages of CaspNets are preserved.
>
>
> Q: The ECCV paper.
> A: Thanks for pointing this out. This paper is definitely related to our paper. We will compare it with our paper.

---

### Official Review · AnonReviewer2 · 2018-10-28
**Reinvention of what is already proposed by Sabour et al.**

**Rating:** 3
**Confidence:** 5

**Review:**

The paper deals with the idea to generalize the CapsNet architecture from Sabour. Under generalization the authors mean, to define a routing procedure without an iteration parameter.

In general, your paper has a good length, is well explained and good organized. To be honest, I don’t like your writing style. It seems to be a bit too casual and not formal enough for a scientific work. Additionally, note that:
-	You are not staring with Fig. 1 in the introduction…the counting should start by one.
-	AC-GAN is the abbreviation for?
-	Eq. 2 is outside the page space.

I have several concerns about the contribution. My major concern is that it isn’t new in general. If I break down your method, it is just the basic Dynamic Routing procedure with:
-	the number of iterations defined to be one;
-	trainable initial routing coefficients;
-	no softmax normalization over routing coefficients.
The usage of trainable initial routing coefficients was already mentioned by Sabour. Thus, the only thing which is new in your method is that you skip the normalization and I’m not sure that this has a positive effect on the process.

Minor concerns/minor mistakes:
1.	You mentioned that the code is public available. Where is the link to a respective repository?
2.	Page 1: “[…] so it makes sense to believe that CapsNet has a better generalization ability.” Compared to what?
3.	Page 2: “The routing iterations is a meta-parameter that needs to be set manually which limits the scalability of CapsNet […].” Why it should limit the scalability? It has no effect on the model size, etc. It’s just a parameter which has to be defined.
4.	Page 3: Are you sure that a linear transformation of a hyper-cube is defined in that way in general?
5.	Page 4: What is T_k?
6.	Page 7: “Hinton et al. (2018) claimed that CapsNets […]” Are you aware that Hinton worked on Matrix Capsules and not on the CapsNet architecture of Sabour?
7.	Page 8: How you can guarantee the convergence of your method? Moreover, the convergence to what?
8.	Could you add some histograms plots of your c_ij values after the training?
9.	Why are your performance values so bad compared to CapsNet and Matrix Capsules?
10.	Could you add to your tables the inference, training times? If you remove the iteration parameter I would assume that your method should be faster, or?
11.	Is the parameter lambda in Eq. 2 the same as in Eq. 6? How you tune that parameter?

---

> ### Author Response · Authors · 2018-11-06
> **Explain the routing procedure in G-CapsNet**
>
> Q: The routing procedure
> A: It seems like you misunderstood the routing procedure of G-CapsNet. We do not use dynamic routing at all. The routing procedure is for combing the capsules in the lower layer. Dynamic routing is based on the similarity between two capsules while our way is training the coefficients for each capsule. The advantage of G-CapsNet is that all the routing coefficients are guaranteed to be (locally) optimal (as Equation 2 shows, the routing procedure is part of the optimization) while the CapsNets (Sabour et al., 2017) can not. We can get the routing coefficients just like we acquire the normal parameters of a neural network.
>
> Q: Code
> A: https://github.com/chenzhenhua986/CAFFE-CapsNet
>
> Q: Page 1: “[…] so it makes sense to believe that CapsNet has a better generalization ability.” Compared to what?
> This statement is an assumption, that is why we apply an experiment in section 4.1.2. We use a GAN to generate artificial images to test whether G-CapsNets generalize better.
>
> Q: Page 2: “The routing iterations is a meta-parameter that needs to be set manually which limits the scalability of CapsNet […].” Why it should limit the scalability? It has no effect on the model size, etc. It’s just a parameter which has to be defined
> A: CapsNets (Sabour et al., 2017) can not guarantee the optimal routing coefficients are optimal and finding the best routing numbers for a deep CapsNet is computationally expensive. For example, in the paper (Sabour et al., 2017), the routing number has to be set as 3. According to our experiment, a routing number that is smaller than 3 or larger than 3 would cause degradation of performance.  For a 10-layer CapsNet, assume we have to try 3 routing numbers for each layer, then totally we have to try 3^10 times to find the best routing number assignment. That is why the scalability and efficiency of CapsNets are questioned. For G-CapsNets, we can compute the routing coefficients just like computing the normal parameters of a neural network, and the (local) optimality can be guaranteed.
>
> Q:  Page 3: Are you sure that a linear transformation of a hyper-cube is defined in that way in general?
> A: Thanks for pointing this out. I am not sure if this is a general way, we will modify our paper and explain that this is one possible way.
>
> Q: Page 4: What is T_k?
> A: It comes from (Sabour et al., 2017), signifies positive samples and negative samples. We will add an explanation in our paper.
>
> Q: Page 7: “Hinton et al. (2018) claimed that CapsNets […]” Are you aware that Hinton worked on Matrix Capsules and not on the CapsNet architecture of Sabour?
> A: Yes, we are aware of that. Could you please illustrate your question? I am not sure what you were asking.
>
> Q: Page 8: How you can guarantee the convergence of your method? Moreover, the convergence to what?
> A: We can guarantee the convergence because we incorporate the routing procedure into the whole optimization process, as Equation 2 shows. We treat the routing coefficients the same as other parameters in a neural network. Thus, these routing coefficients are guaranteed to converge to (locally) the optimal points.
>
> Q: Could you add some histograms plots of your c_ij values after the training?
> Thanks for the suggestion, we will add the histograms of our routing coefficients.
>
>
> Q: Why are your performance values so bad compared to CapsNet and Matrix Capsules?
> A: There are two possible reasons that our accuracy is lower than the original one. 1). We do not use any data-augmentation technique while the original paper (Sabour et al., 2017) adopts 2-pixel shifting, "Training is performed on 28 × 28 MNIST (LeCun et al. [1998]) images that have been shifted by up to 2 pixels in each direction with zero padding." 2). We use different frameworks. We use Caffe (https://github.com/chenzhenhua986/CAFFE-CapsNet) while the original paper uses TensorFlow (https://github.com/Sarasra/models/tree/master/research/capsules).
> We are not trying to get better performance than CapsNet (Sabour et al., 2017), our primary contribution is providing a scalable version of CapsNet, at the same time, all other advantages (for example, achieving better performance with fewer parameters) of CapsNets are preserved.
>
> Q: Could you add to your tables the inference, training times? If you remove the iteration parameter I would assume that your method should be faster, or?
> A: Sorry we did not mention that in our paper. We train all our models for 10k iterations. As for the speed, assume that calculating the routing coefficients with dynamic routing or EM routing procedure cost the same time as G-CapsNet, G-CapsNets should be faster since we need only one step.
>
> Q: Is the parameter lambda in Eq. 2 the same as in Eq. 6? How you tune that parameter?
> A: Thanks for pointing that out. They are different. The lambda in Eq. 2 is a weight that balances the loss and the regularizer while the lambda in Eq. 6 balances the loss between positive samples and negative samples.

---

> > ### Comment · AnonReviewer2 · 2018-11-27
> > **Routing procedure**
> >
> > About your proposed procedure:
> >
> > If you carefully read the paper of Sabour you will observe that your proposed algorithm is equivalent to what they described. Just define the number of routing steps in Dynamic Routing to one (so just one step in the for loop) and define the initial routing coefficients as trainable parameters. That the initial routing coefficients can be trainable is already described by Sabour. In the end, the only thing that you have changed is that you removed the softmax normalization of routing parameters. It's obvious if you write down your method as pseudo-code.
> >
> > Please correct me again if I'm wrong...

---

> > > ### Author Response · Authors · 2018-11-28
> > > **Dynamic routing has a conflict with trainable routing procedure.**
> > >
> > > Thanks for the comments.
> > >
> > > This is the official code of CapsNet: https://github.com/Sarasra/models/tree/master/research/capsules.
> > > If I am correct, your idea is first setting the routing number as one, then initializing the trouting coefficients as trainable parameters. As the official code shows, the dynamic routing procedure is always executed even when the routing number is one. On the one hand, training the coupling coefficients is based back-propagation (local optimal is guaranteed). On the other hand, dynamic routing is a heuristic procedure which is based on the similarity between capsules in the adjacent layers ( local optimal is not guaranteed). These are two independent ways of calculating the routing coefficients, I can not see how they can work together.

---

### Official Review · AnonReviewer3 · 2018-11-05
**Replacing dynamic routing with trainable layers is interesting, but the contributions of the paper are not very clear.**

**Rating:** 5
**Confidence:** 3

**Review:**

This paper proposes to replace the dynamic routing layer of the original capsule networks with a trainable neural network layer. The idea is interesting. However, the following problems concern me.

The model is named as “generalized capsule networks” but it is not very clear what “generalized” means here. Does making the dynamic routing layer trainable generalize capsule?

The contributions of the paper will be clearer if further comparison is provided. The model is proposed based on capsule nets but it lacks comparison between the proposed model and the original capsule nets. For example, the experiments did not include the original CapsNet models (Sabour et al., 2017 or Hinton et al., 2018), which, if performed, would help understand the differences/advantages of the proposed models.

The major modification made on capsule nets is on the dynamic routing layer. In order to incorporate routing into the whole trainable process, this paper incorporate the coefficients c_{ij} to the model parameters. It is not clear if the proposed model constrains c_{ij} are further constrained, e.g., (as in the original capsule nets) to sum up to 1 along the dimension i?

In general, the paper is well structured and easy to follow.

---

> ### Author Response · Authors · 2018-11-06
> **Our primary contribution is providing a scalable version of CapsNet that can guarantee convergence**
>
>
> Q: The model is named as “generalized capsule networks” but it is not very clear what “generalized” means here. Does making the dynamic routing layer trainable generalize capsule?
> A: By "generalized", we mean we can train a CapsNet just like training a standard neural network. The routing coefficients in the original CapsNet (Sabour et al., 2017) have to be acquired by applying for the routing number (e.g., routing number equals 3) of iterations for each layer and these routing coefficients cannot be guaranteed to be optimal.  For example, in the paper (Sabour et al., 2017), the routing number has to be set as 3. A routing number that is smaller than 3 or larger than 3 would cause degradation of performance.  For a 10-layer CapsNet, assume we have to try 3 routing numbers for each layer, then totally we have to try 3^10 times to find the best routing number assignment. That is why the scalability and efficiency of CapsNets are questioned. For G-CapsNets, we can compute the routing coefficients just like computing the normal parameters of a neural network, and the (local) optimality can be guaranteed. Thus we can build a deep CapsNet (e.g., 100-layer CapsNet) without worrying about how to choose the best routing numbers for each layer.
>
>
> Q: The contributions of the paper will be clearer if further comparison is provided. The model is proposed based on capsule nets but it lacks comparison between the proposed model and the original capsule nets. For example, the experiments did not include the original CapsNet models (Sabour et al., 2017 or Hinton et al., 2018), which, if performed, would help understand the differences/advantages of the proposed models.
> A: That is a good suggestion. We will add the comparison with the original CapsNet.
>
> Q: The major modification made on capsule nets is on the dynamic routing layer. In order to incorporate routing into the whole trainable process, this paper incorporate the coefficients c_{ij} to the model parameters. It is not clear if the proposed model constrains c_{ij} are further constrained, e.g., (as in the original capsule nets) to sum up to 1 along the dimension i?
>
> A: A short answer to your question is the routing coefficients are not necessarily, to sum up to 1. As Equation 2 shows, the routing coefficients are constrained to two terms. One is the loss term, and the other one is the regularizer.  In other words, G-CapsNets choose whatever routing coefficients that can best fit the loss function. Let me specify it.  The routing procedure is for combing the capsules in the lower layer. Dynamic or EM routing is based on the similarity between two capsules while our method is training the coefficients for each capsule. The advantage of G-CapsNet is that all the routing coefficients are guaranteed to be (locally) optimal (as Equation 2 shows, the routing procedure is part of the optimization) while the CapsNets (Sabour et al., 2017) can not.

---

### Author Response · Authors · 2018-11-06
**Code is available now**

https://github.com/chenzhenhua986/CAFFE-CapsNet

---

### Meta-Review · Area_Chair1 · 2018-12-13
**Promising as a direction of research but reviewers have several concerns with the paper**

**Confidence:** 5
**Recommendation:** Reject

**Metareview:**

The paper proposes to replace dynamic routing in Capsule networks with a trainable layer that produces routing coefficients. The goal is to improve their scalability. This is promising as a research direction but reviewers have raised several concerns about unclear contributions and lack of a thorough evaluation of the approach. There is also a recent relevant work pointed out by Reviewer 1 that should be discussed. Given these concerns, the paper is not suitable for publication in its current form, however I encourage the authors to use reviewers' comments for improving the paper and resubmit in next venues.